# Multidrug-Resistant *Klebsiella pneumoniae* Causing Severe Infections in the Neuro-ICU

**DOI:** 10.3390/antibiotics10080979

**Published:** 2021-08-13

**Authors:** Nadezhda K. Fursova, Evgenii I. Astashkin, Olga N. Ershova, Irina A. Aleksandrova, Ivan A. Savin, Tatiana S. Novikova, Galina N. Fedyukina, Angelina A. Kislichkina, Mikhail V. Fursov, Ekaterina S. Kuzina, Sergei F. Biketov, Ivan A. Dyatlov

**Affiliations:** 1Department of Molecular Microbiology, State Research Center for Applied Microbiology and Biotechnology, Territory “Kvartal A”, 142279 Obolensk, Russia; info@obolensk.org (E.I.A.); pozitifka.15@yandex.ru (T.S.N.); 2Department of Clinical Epidemiology, National Medical Research Center of Neurosurgery Named after Academician N.N. Burdenko, 125047 Moscow, Russia; oershova@nsi.ru (O.N.E.); ialexandrova@nsi.ru (I.A.A.); info@nsi.ru (I.A.S.); 3Department of Immunobiochemistry of Pathogenic Microorganisms, State Research Center for Applied Microbiology and Biotechnology, Territory “Kvartal A”, 142279 Obolensk, Russia; galafed@mail.ru (G.N.F.); biketov@obolensk.org (S.F.B.); 4Department of Culture Collection, State Research Center for Applied Microbiology and Biotechnology, Territory “Kvartal A”, 142279 Obolensk, Russia; angelinakislichkina@yandex.ru; 5Department of Training and Improvement of Specialists, State Research Center for Applied Microbiology and Biotechnology, Territory “Kvartal A”, 142279 Obolensk, Russia; mikhail.fursov88@gmail.com (M.V.F.); e.leonova@mail.ru (E.S.K.); 6Department of Administration, State Research Center for Applied Microbiology and Biotechnology, Territory “Kvartal A”, 142279 Obolensk, Russia; dyatlov@obolensk.org

**Keywords:** *Klebsiella pneumoniae*, healthcare-associated infections, microbial drug resistance, sequence type, capsular type, whole genome sequencing

## Abstract

The purpose of this study was the identification of genetic lineages and antimicrobial resistance (AMR) and virulence genes in *Klebsiella pneumoniae* isolates associated with severe infections in the neuro-ICU. Susceptibility to antimicrobials was determined using the Vitek-2 instrument. AMR and virulence genes, sequence types (STs), and capsular types were identified by PCR. Whole-genome sequencing was conducted on the Illumina MiSeq platform. It was shown that *K. pneumoniae* isolates of ST14^K2^, ST23^K57^, ST39^K23^, ST76^K23^, ST86^K2^, ST218^K57^, ST219^KL125/114^, ST268^K20^, and ST2674^K47^ caused severe systemic infections, including ST14^K2^, ST39^K23^, and ST268^K20^ that were associated with fatal incomes. Moreover, eight isolates of ST395^K2^ and ST307^KL102/149/155^ were associated with manifestations of vasculitis and microcirculation disorders. Another 12 *K. pneumoniae* isolates of ST395^K2,KL39^, ST307^KL102/149/155^, and ST147^K14/64^ were collected from patients without severe systemic infections. Major isolates (*n* = 38) were XDR and MDR. Beta-lactamase genes were identified: *bla*_SHV_ (*n* = 41), *bla*_CTX-M_ (*n* = 28), *bla*_TEM_ (*n* = 21), *bla*_OXA-48_ (*n* = 21), *bla*_NDM_ (*n* = 1), and *bla*_KPC_ (*n* = 1). The prevalent virulence genes were *wabG* (*n* = 41), *fimH* (*n* = 41), *allS* (*n* = 41), and *uge* (*n* = 34), and rarer, detected only in the genomes of the isolates causing severe systemic infections—*rmpA* (*n* = 8), *kfu* (*n* = 6), *iroN* (*n* = 5), and *iroD* (*n* = 5) indicating high potential of the isolates for hypervirulence.

## 1. Introduction

Healthcare-associated infections (HAI) have posed a huge medical burden to public health worldwide. *Klebsiella pneumoniae* is one of the clinically significant nosocomial pathogens causing broad spectra of diseases and showing increasingly frequent acquisition of resistance to antibiotics including in intensive care units (ICU) [1]. Today, according to the BIGSDB Institute Pasteur database (https://bigsdb.pasteur.fr/ access date 8 August 2021), 5797 *K. pneumoniae* sequence types and 711 capsular types have been discovered. *K. pneumoniae* causes different infections in ICU patients associated with high mortality rates, including bloodstream infection, pulmonary infection, and healthcare-associated ventriculitis, meningitis, and urinary tract infection [2,3]. It was reported from China, Spain, and Taiwan that *K. pneumoniae* was associated with leukocytoclastic vasculitis and microcirculation disorders, but there is no data on the relationship of such a clinical manifestation with any *K. pneumoniae* genetic lineages [4,5,6]. 

Current data suggest that genetic determinants associated with antimicrobial resistance (AMR) and virulence are specific for distinct genetic lineages of *K. pneumoniae* [7]. Classical *K. pneumoniae* (cKP) are the common healthcare-associated pathogens causing nosocomial infections in immunocompromised patients and characterized as a great possibility to accumulate resistance mechanisms to antimicrobials—multidrug-resistant (MDR), extensive drug-resistant (XDR), and pan drug-resistant (PDR) strains have been described [8]. Carbapenems are the last antibiotics available to control *K. pneumoniae* infection. However, to date, many mechanisms involved in carbapenem resistance in *K. pneumoniae* have already been described, including carbapenemase production. Strains producing carbapenemases of functional class A (KPC), class B (NDM), and class D (OXA-48), and co-producing more than one type of carbapenemase are increasingly reported [9].

For hypervirulent *K. pneumoniae* (hvKP)*,* many critical virulence factors have been discovered, among them *rmpA* (regulator of mucoid phenotype), aerobactin (an iron siderophore), *kfu* (an iron uptake system), *allS* (associated with allantoin metabolism), and K1/K2 capsules [10]. The global spread of ‘convergent’ *K. pneumoniae* strains that combines the pathogenic potentials of cKP and hvKP has occurred since 2010. Three types of the mechanism for the emergence of ‘convergent’ clones have been identified: (i) cKP acquiring a hypervirulence plasmid; (ii) hvKP acquiring a carbapenem-resistant plasmid; and (iii) *K. pneumoniae* acquiring both a carbapenem resistance and hypervirulence hybrid plasmid. These strains are at high risk to disseminate and have the potential to cause severe infections [9].

This study aimed to examine the phenotypes, genotypes, and genetic relatedness of *K. pneumoniae* isolates collected from the patients of neuro-ICU and evaluated the sequence types associated with severe infections between 2017 and 2019. 

## 2. Results

### 2.1. Patients and Bacterial Strains

During the period from Oct. 2017 to Jan. 2019, the following incidence of infections was detected in neurosurgery ICU: 8.4 infections of the central nervous system per 100 patients, 2.7/100 of bloodstream infections, 26.3/100 of ventilator-associated pneumonia, and 23.6/100 of urinary tract infections. *K. pneumoniae* accounted for 33, 31, 23, and 25% among the agents of named infections, correspondingly [3]. The subject of this study was 41 resistant-to-antimicrobials *K. pneumoniae* isolates collected from 20 patients with severe postoperative infections (Table 1). 

Depending on clinical manifestations, isolates were combined into two groups. Group A included 29 isolates collected from 13 patients (D, J, Q, P, M, S, L, K, N, R, A, I, and F) with the pronounced systemic inflammatory response (SIRS). Symptoms had been reported for the patients: fever >38 °C, leukocytosis, increased markers of systemic inflammation, multiple hemorrhages, thrombosis, or high risk of thromboembolism. Clinical forms had been detected: meningitis after the neurosurgical intervention, pansinusitis, pneumoniae, sepsis and septic shock, ventriculitis, and brain abscess. In this group, seven isolates (B-548/18, B-784/18, B-1154/18, B-1618/18, B-2625/18, B-14/19, and B-21/19) were associated with fatal incomes collected from the patients D, Q, and S died due to increased symptoms of inflammation, sepsis, septic shock, meningoencephalitis associated with *K. pneumoniae* infection. Other eight isolates (B-3002K/17, B-3060K/17, B-2016K/17, B-3299/17, B-1040/18-1, B-792/18, B-853/18-1, and B-1214/18-2) of Group A were associated with manifestations of vasculitis and microcirculation disorders (patients A, I, and F). Major isolates of Group A (17/29) were collected from the blood and cerebrospinal fluid, less (7/29)—from the endotracheal aspirate.

Group B consists of 12 strains isolated from the patients without increasing markers of systemic inflammation and septic reaction. Major isolates of Group B were from the endotracheal aspirate and urine (10/12) and the rest from the nervous system. One patient in this group died due to the underlying disease, videlicet multiple metastases of kidney cancer in the brain (Table 1 and Table 2). It should be noted that isolates of Group A were obtained from 2–3 body sites of one patient: three isolates—from blood, urine, and cerebrospinal fluid of the patient D; three isolates—from endotracheal aspirate and cerebrospinal fluid of patient S; and three isolates—from blood and cerebrospinal fluid of the patient L (Table 2).

### 2.2. Susceptibility to Antimicrobials

According to Magiorakos et al. [11] criteria, 28 isolates were attributed to the XDR category (resistant to 6–7 functional groups of antimicrobials), 10 isolates to the MDR category (resistant to 3–4 functional groups), and 3 isolates to the R category (resistant to ampicillin only). XDR isolates were attributed to Groups A (*n* = 18) and B (*n* = 10); MDR isolates—to the Groups A (*n* = 8) and B (*n* = 2); R isolates—only to Group A (*n* = 3). All isolates were resistant to beta-lactams, 34 isolates—to nitrofurans, 33 isolates—to fluoroquinolones, 29 isolates—to chloramphenicol, 29 isolates—to sulfonamides, 27 isolates—to aminoglycosides, 26 isolates—to tetracyclines. Major isolates were susceptible to amikacin (*n* = 33) and imipenem (*n* = 27), all isolates were susceptible to colistin (Figure 1, Appendix A).

### 2.3. Beta-Lactamase Genes and Integrons

Total 113 beta-lactamase genes were identified in *K. pneumoniae* isolates, including *bla*_SHV_ (*n* = 41), *bla*_CTX-M_ (*n* = 28), *bla*_TEM_ (*n* = 21), *bla*_OXA-48_ (*n* = 21), *bla*_NDM_ (*n* = 1), and *bla*_KPC_ (*n* = 1). The *bla*_VIM_ and *bla*_IMP_ genes were not detected. Number of beta-lactamase genes per strain varied from one to five. Single *bla*_SHV_ gene was detected in 4 strains; two genes (*bla*_SHV_ + *bla*_CTX-M_) or (*bla*_SHV_ + *bla*_OXA-48_)—in 4 and 9 strains, respectively; three genes (*bla*_SHV_ + *bla*_CTX-M_ + *bla*_OXA-48_) or (*bla*_SHV_ + *bla*_CTX-M_ + *bla*_TEM_)—in 3 and 11 strains, respectively; four genes (*bla*_SHV_ + *bla*_CTX-M_ + *bla*_TEM_ + *bla*_NDM_) or (*bla*_SHV_ + *bla*_CTX-M_ + *bla*_TEM_ + *bla*_OXA-48_)—in 1 and 8 strains, respectively; five genes (*bla*_SHV_ + *bla*_CTX-M_ + *bla*_TEM_ + *bla*_OXA-48_ + *bla*_KPC_) were identified in one strain. Moreover, 10 strains carried class 1 integrons, and one strain carried two integrons, class 1 and class 2 simultaneously (Table 3).

### 2.4. Virulence Genes

The prevalent virulence genes detected in 41 *K. pneumoniae* clinical isolates were *wabG* (*n* = 41), *fimH* (*n* = 41), *allS* (*n* = 41), and *uge* (*n* = 34). Other virulence genes were rare and attributed only to the Group A: *rmpA* (*n* = 8), *kfu* (*n* = 6), *iroN* (*n* = 5), and *iroD* (*n* = 5). Eight combinations of 3 to 8 virulence genes were identified. Combination of three genes (*wabG* + *fimH* + *allS*) was detected in three isolates; combinations of four genes (*uge* + *wabG* + *fimH* + *allS*) and (*rmpA* + *wabG* + *fimH* + *allS*)—in 24 and 2 isolates, respectively; combinations of five genes (*uge* + *wabG* + *kfu* + *fimH* + *allS*) and (*rmpA* + *uge* + *wabG* + *fimH* + *allS*)—in 6 and 1 isolates, respectively; combinations of six genes (*rmpA* + *iroN* + *iroD* + *wabG* + *fimH* + *allS*)—in 2 isolates; combinations of seven genes (*rmpA* + *iroN* + *iroD* + *uge* + *wabG* + *fimH* + *allS*)—in 3 isolates (Table 3).

### 2.5. Sequence Types and Capsular Types

It was found that *K. pneumoniae* clinical isolates belonged to 12 sequence types: ST14 (*n* = 3), ST23 (*n* = 5), ST39 (*n* = 3), ST76 (*n* = 1), ST86 (*n* = 2), ST147 (*n* = 1), ST218 (*n* = 2), ST219 (*n* = 3), ST268 (*n* = 1), ST307 (*n* = 3), ST395 (*n* = 16) and ST2674 (*n* = 1). Nine sequence types (ST14, ST23, ST39, ST76, ST86, ST218, ST219, ST268, and ST2674) were identified only in Group A. One sequence type (ST147) was obtained only in Group B. Two sequence types (ST395 and ST307) were identified both in Group A (8 isolates associated with systemic manifestations of vasculitis and microcirculation disorders) and in Group B.

All studied *K. pneumoniae* isolates were attributed to nine capsular types: K2 (*n* = 20), K20 (*n* = 1), K23 (*n* = 4), KL39 (*n* = 1), K47 (*n* = 1), K57 (*n* = 7), K14/64 (*n* = 1), K102/149/155 (*n* = 3), and K125/114 (*n* = 3). It was interesting in that *K. pneumoniae* isolates of capsular type K2 belonged to three sequence types (ST14, ST86, and ST395), isolates of capsular type K23—to two sequence types (ST39 and ST76), and isolates of capsular type K57—to two sequence types (ST23 and ST218). The remaining capsular types were associated with only one sequence type: K20—ST268, KL39—ST395, K47—ST2674, K14/64—ST147, K102/149/155—ST307, and K125/114—ST219 (Table 3).

### 2.6. Phylogenetic Analysis

The phylogenetic tree was constructed on the base of combined gene sequences of MLST profiles; two clusters were revealed. Cluster I consisted of one ST147 referring to one isolate not associated with severe manifestations of systemic infections (Group B). Cluster II included two sub-clusters: IIa consisting of one ST86 associated with a pronounced systemic inflammatory response (Group A); IIb consisting of 10 genetic lineages including 8 sequence types (ST14, ST23, ST39, ST76, ST218, ST219, ST268, and ST2674) referred to the strains of Group A, and 2 sequence types (ST307 and ST396) including isolates both Group A and Group B (Figure 2).

### 2.7. Whole-Genome Sequencing

Whole-genome sequencing was done for nine isolates including eight isolates of Group A belonging to sequence types/capsular types ST2674/K47, ST23/K57, ST39/K23, ST219/K125, ST218/K57, ST76/K23, ST86/K2, and ST307/K102 and one isolate of Group B belonging to ST395/K39. From 86 to 164 contigs for each genome were obtained, the genome size ranged from 5.42 to 5.86 Mb, with the GC content ranging from 55.1 to 57.9%, and the number of genes from 5121 to 5845. All genomes carried 1–6 beta-lactamase genes including *bla*_SHV_, *bla*_TEM_, *bla*_CTX-M_, *bla*_OXA_, and *bla*_KPC_ types. Six alleles were identified of the *bla*_SHV_ gene (26, 28, 33, 40, 59, and 182), two alleles of *bla*_OXA_ genes (1 and 48), and other beta-lactamase genes were presented as only one allele (*bla*_TEM-1B_, *bla*_CTX-M-15_, and *bla*_KPC-2_). The genetic environments of carbapenemase genes *bla*_KPC-2_ and *bla*_OXA-48_ and cefalosporinase gene *bla*_CTX-M-15_ were similar to previous descriptions of these genes. The gene coding of a putative transposition helper protein ISKpn7 was identified upstream *bla*_KPC-2_ gene, and ISKpn6 transposase gene downstream, that was similar to the genetic structure in the plasmid pBK32179 (JX430448). The environment for *bla*_OXA-48_ genes contained upstream, the transcriptional regulator LysR gene, and the transposase *tir* gene downstream, as in the plasmid pOXA-48 (JN626286). The surrounding genetic conditions for *bla*_CTX-M-15_ genes in clinical isolates were likewise similar to the plasmid pKp848CTX (NC_024992): ISEcp1 family transposase upstream and gene coding WbuC family cupin fold metalloprotein downstream. Aminoglycoside modifying enzyme genes were identified in all but one isolate: 1–3 genes per isolate (*aac, aad, ant, aph, arm,* and *rmt*). All genomes carried the *fosA* gene conferring resistance to fosfomycin. Phenicol resistance genes (4 alleles of *cat* gene) were detected in seven genomes. Six isolates carried 2–4 quinolone resistance genes (*aac*, *qnr*, *oqx*, and *qep*). Two alleles of *sul* gene (sulfonamide resistance) were identified in six genomes, *tet* gene (tetracycline resistance) in 5 genomes, four alleles of *dfrA* gene (trimethoprim resistance) in 4 genomes, and *mph* gene (macrolide resistance) in only one genome. The genes encoding efflux pumps were revealed in all isolates, seven of them carried 10 genes, two isolates—9 genes. Moreover, all genomes carried 1–4 genes encoding heavy metal resistance (to copper, lead, silver, arsenic, and tellurium) (Table 4).

Analysis of virulence genetic determinants revealed *mrk* gene coding type 3 adhesine—in all 9 genomes, *irp* gene of yersiniabactin biosynthesis and *ybt* gene of yersiniabactin transcriptional regulator—in 7 genomes, *fyu* gene of siderophore yersiniabactin receptor—in 6 genomes, *iut* gene of ferric aerobactin receptor—in 6 genomes, *iuc* gene of aerobactin siderophore synthesis—in 5 genomes, *kvg* gene of capsular polysaccharide synthesis regulator—in 2 genomes. Generally, each genome contained 1–7 virulence genes. Plasmids of eight incompatibility groups (IncC, IncFIA, IncFIB, IncFII, IncHI1B, IncM1, IncM2, and IncR) were identified in the *K. pneumoniae* isolate genomes, specifically 1–3 plasmids per genome. Molecular systems protecting bacteria from the foreign DNA, Type I Restriction-Modification systems, were detected in four genomes, Type II systems in all nine genomes, but CRISPR-Cas systems were not detected in the genomes (Table 4).

## 3. Discussion

*K. pneumoniae* isolates that caused severe postoperative infections in patients of neuro-ICU between 2017 and 2019 were divided into two groups, A and B, depending on observed clinical manifestations: associated and not associated with the pronounced systemic inflammatory response. Previously, the association of *K. pneumoniae* with severe systemic infections was described as the modern trend for ICU; for example, *K. pneumoniae* were the most frequent infecting species (47%) determined meningitis/encephalitis and 30-day mortality rates, 15% in post-neurosurgical patients [12]. In our study, as in the reports from other countries, the overwhelming majority of *K. pneumoniae* isolates obtained from neurosurgery patients were MDR and XDR [13]. Moreover, MDR and XDR *K. pneumoniae* isolates in our study were associated with severe systemic infections, including vasculitis and microcirculation disorders. It was reported previously that *K. pneumoniae* was associated with leukocytoclastic vasculitis [4,5,6], rapidly progressive retinal vasculitis [14], and acute vasculitis at respiratory infection [15]. In this study, we first identified *K. pneumoniae* of ST395 and ST307 as bacterial pathogen associated with vasculitis and microcirculation disorders.

A total of twelve sequence types and nine capsular types of *K. pneumoniae* were identified in this study, which possibly reflects a continuous influx of new genetic lineages into neuro-ICU from other hospitals and regions. The prevalent ST in the ICU was ST395 (16/41 isolates), which is similarly rated to that previously published in the studies from Poland, France, Italy, and Russia [16,17,18,19,20]. Nine STs in our study (ST14, ST23, ST39, ST76, ST86, ST218, ST219, ST268, and ST2674) were identified only for the isolates that caused severe bloodstream infections. Previously *K. pneumoniae* of ST23, ST86, ST76, and ST218 were described as a hypervirulent pathogen causing bacteremia, sepsis, and liver abscess in India, France, China, Taiwan, and Russia [21,22,23,24,25]. Three STs (ST14, ST39, and ST268) in our study were associated with fatal outcomes. These STs have been reported previously as the agents of severe bloodstream infections in ICU and surgery wards in other countries [26,27]. Two sequence types, ST219, and ST2674 were first identified in this study as the agent of severe sepsis in the patients of ICU. Recently, the ST219 was reported for the environmental MDR *K. pneumoniae* strains collected from hospital wastewater in Southern Romania [28]; the ST2674 was identified for the environmental isolate from Pakistan (https://bigsdb.pasteur.fr/cgi-bin/bigsdb/bigsdb.pl?page=info&db=pubmlst_klebsiella_isolates&id=5256 access date 8 August 2021). 

The high prevalence of polyresistant isolates in our study was associated with antimicrobial resistance genes. The isolates of Group A carried ESBL genes *bla*_CTX-M_ (22/29), carbapenemase genes *bla*_OXA-48_ (12/29), and *bla*_KPC-2_ (1/29), which correspond to the reports from Greece and China [27,29,30]. Moreover, four isolates in Group A carrying the only *bla*_SHV_ were identified (ST218 and ST86). These isolates, as well as four additional isolates of ST268, ST23, and ST76, carried virulence gene *rmpA* coding regulator of hypermucoid phenotype. It was reported previously that overexpression of *rmpA* could enhance the virulence of *K. pneumoniae* isolates in the mouse model [31]. Virulence genes *iroN* and *iroD* associated with utilization of trivalent iron were detected in *K. pneumoniae* of ST23, ST218, ST76, ST86, and ST268 (isolates associated with severe bloodstream infections) which agrees with reports from other countries [22,23,29]. It is known that the majority of pathogenic bacteria including *K. pneumoniae* possess the iron-acquisition system with a higher affinity for iron than the host, which serves as one of the strategies for increasing bacterial virulence [32]. The *kfu* gene coding iron uptake system was identified in *K. pneumoniae* of ST14 and ST219; the latter is a novel genetic lineage carrying the *kfu* gene [33,34]. *Kfu* was shown to be a potential virulence factor in the intragastrical murine model, which indicates that *kfu* might contribute to intestinal colonization [35]. Therefore, the identified virulence genes indicate the high potential of studied isolates for hypervirulence.

The isolates that were not associated with severe manifestations of systemic infections in our study were attributed to ST395, ST307, and ST147. Among them, the isolates of ST395 carrying the *bla*_OXA-48_ carbapenemase gene were prevalent. Such *K. pneumoniae* strains were reported earlier from Hungary and Russia [36,37]. In our study, one isolate of ST147 carried the *bla*_NDM-1_ carbapenemase gene, and the same strains have been described worldwide [38,39]. It should be noted that ST307 and ST147 have been estimated as *K. pneumoniae* High-Risk Clones (HRC) because of worldwide distribution, ability to cause serious infections, and association with polyresistance [40].

The whole-genome study of selected *K. pneumoniae* isolates belonged to nine sequence types, ST23, ST39, ST76, ST86, ST218, ST219, ST307, ST395, and ST2674, showing the great diversity of these isolates in the combination of virulence genes, antimicrobial resistance genes, heavy metal resistance, and plasmids. Analysis of their resistomes showed that the genes of beta-lactamases *bla*_CTX-M-15_, *bla*_TEM-1B_, *bla*_NDM-1_, *bla*_OXA-48_, *bla*_KPC-2_, and *bla*_OXA-1_ are represented by a single allele. These alleles were recently reported for *K. pneumoniae* isolated in Russia [41]. On the contrary, six alleles were identified of *bla*_SHV_-type genes, which were not reported in Russia before our study: *bla*_SHV-26_, *bla*_SHV-28_, *bla*_SHV-33_, *bla*_SHV-40_, *bla*_SHV-59_, and *bla*_SHV-182_. In general, all isolates in our work carried *bla*_SHV_ genes, 68% isolates—*bla*_CTX-M_ genes, 51%—*bla*_TEM_ genes, 51%—*bla*_OXA-48_ genes, but only one isolate carried *bla*_NDM-1_ gene, and only one isolate *bla*_KPC-2_ gene. A similar representation of beta-lactamase genes was reported from the European countries and Russia [41,42]. Major isolates in our study were susceptible to amikacin and imipenem, which is consistent with recently published data from Saudi Arabia and Indonesia [43,44]. Interestingly, the *rmtB* gene encoding 16S rRNA methylase providing resistance to aminoglycosides in the *K. pneumoniae* ST23 isolate was detected in this study for the first time. This gene was reported earlier for ST258 and ST16 of KPC-producing *K. pneumoniae* [45]. Moreover, *armA* gene coding 16S rRNA methyltransferase was detected in *K. pneumoniae* of ST395 in this study; recently, this gene was detected in *K. pneumoniae* ST23, ST2502, and ST11 in Italy, Spain, and China, respectively [46,47].

The virulence determinants detected in the genomes were the following: mannose-resistant *Klebsiella*-like (*mrk*) hemagglutinin gene critical for *K pneumoniae* biofilm development in all 9 genomes; aerobactin siderophore locus (*iuc*, *iut*) in 6 genomes; regulator of capsular polysaccharide synthesis (*kvg*) in 2 isolates. A similar distribution of virulence genes was reported recently from Brazil [48]. The yersiniabactin locus (*irp*, *ybt*, and *fyu*) was detected in 7/9 genomes in our study, compared with 40% of *K. pneumoniae* genomes, particularly amongst those associated with invasive infections [49]. This data confirmed the high virulence potential of the studied clinical isolates. The presence in the genomes of the plasmids of different incompatibility groups indicates a possible role of ‘hybrid’ plasmids in the formation of *K. pneumoniae* strains, simultaneously carrying a large number of antibiotic resistance and virulence genes. [9,50].

Future research will focus on studying the structure of plasmids carrying genes for antimicrobial resistance and virulence, as well as the expression of these genes under conditions of different genetic environments and selective pressure of antibiotics. We believe that further study of the microbiology, molecular biology, physiology, and interactions with the host of *K. pneumoniae* will provide important knowledge to control *K. pneumoniae* infection in ICUs.

## 4. Materials and Methods

### 4.1. Bioethical Requirements and Patients

Each patient signed informed voluntary consent to treatment and laboratory examination, under the requirements of the Russian Federation Bioethical Committee. The study did not contain personal data of patients; the clinical bacterial isolates information did not include name, date of birth, address, and disease history. The study was a retrospective observational study. The study was done in the neuro-ICU department in a specialized Neurosurgical Hospital in Moscow, Russia, with 300 beds that care for approximately 8000 patients per year, 95% of whom undergo surgery. The surveillance software was designed in-house and integrated into the hospital’s electronic health record system [51]. Four types of health-associated infections were surveilled: bloodstream, respiratory and urinary tract infections, and healthcare-associated ventriculitis and meningitis [3].

### 4.2. Bacterial Isolates and Growing

*K. pneumoniae* isolates (*n* = 41) were collected from clinical samples (endotracheal aspirate, cerebrospinal fluid, blood, wound exudate, and urine) of the patients (*n* = 20) in the Neuro-ICU in the period from October 2017 to January 2019. Bacterial identification was performed using a Vitek-2 Compact (BioMerieux, Paris, France) and a MALDI-TOF Biotyper (Bruker Daltonics, Bremen, Germany) instruments. Bacterial isolates were grown at 37 °C on Nutrient Medium No. 1 (SRCAMB, Obolensk, Moscow region, Russia), Luria-Bertani broth (Difco Laboratories, Detroit, MI, USA), and Muller-Hinton broth (Himedia, Mumbai, Maharashtra, India). Bacterial isolates were stored in 15% glycerol at minus 80 °C.

### 4.3. Antimicrobial Susceptibility

Susceptibility to 20 antimicrobials (AMs) of 8 functional groups: beta-lactams (ampicillin, ampicillin-sulbactam, cefuroxime, cefoxitin, ceftriaxone, ceftazidime, cefoperazone-sulbactam, cefepime, ertapenem, imipenem), tetracyclines (tetracycline, tigecycline), fluoroquinolones (ciprofloxacin), phenicols (chloramphenicol), aminoglycosides (gentamicin, tobramycin, amikacin), sulfonamides (trimethoprim-sulfamethoxazole), nitrofurans (nitrofurantoin), and polymyxins (colistin) were determined using Vitek-2 Compact (BioMerieux, Paris, France). The results were interpreted according to the European Committee on Antimicrobial Susceptibility Testing, Version 11.0, 2021 (http://www.eucast.org access date 8 August 2021). Reference strains *Escherichia coli* ATCC 25922 and ATCC 35218 were used as quality controls. The isolates were categorized as resistant (R), multidrug-resistant (MDR), or extensively drug-resistant (XDR) according to the criteria proposed by Magiorakos et al. [11].

### 4.4. Detection of Antimicrobial Resistance Genes

Beta-lactamase genes *bla*_SHV_, *bla*_CTX-M_, *bla*_TEM_, *bla*_OXA-48_, *bla*_KPC_, *bla*_VIM_, *bla*_IMP_, and *bla*_NDM_, and class 1 and 2 integrons were detected by PCR using previously described specific primers [52,53,54,55,56,57,58,59,60].

### 4.5. Detection of Virulence Genes and Capsular Type Identification

Eight genes associated with *K. pneumoniae* virulence, *rmpA* (hypermucoid phenotype regulator), *aer* (aerobactin), *kfu* (ferric absorption system), *uge* (uridine diphosphate-galacturonate-4-epimerase), *wabG* (glucosyltransferase), *fimH* (fimbria type I), *allS* and *allR* (allantoin regulon), were detected by PCR using specific primers [35,61,62,63]. The capsular serotypes of the *K. pneumoniae* isolates were determined by *wzi* gene amplification using specific primers [64], sequencings, and allele identification using the Institute Pasteur, Paris, France, BIGS database (https://bigsdb.pasteur.fr/ access date 8 August 2021).

### 4.6. Multilocus Sequence Typing

Sequence types (STs) of *K. pneumoniae* isolates were identified based on allelic profiles of seven housekeeping genes (*rpoB, gapA, mdh, pgi, phoE, infB*, and *tonB*), according to the Institute Pasteur, Paris, France, BIGS database protocol (https://bigsdb.pasteur.fr/klebsiella/primers_used.html access date 8 August 2021). Sequencing of DNA fragments was carried out at the SINTOL Center for collective use (Moscow, Russia). DNA sequences were analyzed using Vector NTI9 (Invitrogen, Carlsbad, CA, USA) and BLAST (https://blast.ncbi.nlm.nih.gov/Blast.cgi access date 8 August 2021).

### 4.7. Phylogenetic Analysis

The phylogenetic tree of *K. pneumoniae* STs was constructed using a web resource NCBI “Blastn” and “Blast Tree View” (https://blast.ncbi.nlm.nih.gov/Blast.cgi?PROGRAM=blastn&PAGE_TYPE=BlastSearch&LINK_LOC=blasthome access date 8 August 2021), based on combined gene sequences of MLST profiles.

### 4.8. Whole-Genome Sequencing 

Whole-genome sequencing was done on the Illumina MiSeq platform using Nextera DNA Library Preparation Kit (Illumina, Carlsbad, CA, USA) and MiSeq Reagent Kits v3 (Illumina, Carlsbad, CA, United States). The obtained single reads were collected into contigs using the SPAdes 3.9.0 software (Petersburg State University, St-Petersburg, Russia). De novo assembled genomes were annotated in the GenBank database (https://github.com/ncbi/pgap access date 8 August 2021). AM resistance genes, STs, and plasmids were identified using the web resources of the Center for Genomic Epidemiology: ResFinder, KmerResistance (90% identity threshold and 10% threshold for depth corr.), MLST, and PlasmidFinder (95% threshold for minimum identity and 60% minimum coverage) (http://www.genomicepidemiology.org/ access date 8 August 2021). Virulence genes, capsular type, and efflux pumps were identified by the Institut Pasteur, Paris, France, BIGS database web-resource of (https://bigsdb.pasteur.fr/ access date 8 August 2021).

## Figures and Tables

**Figure 1 antibiotics-10-00979-f001:**
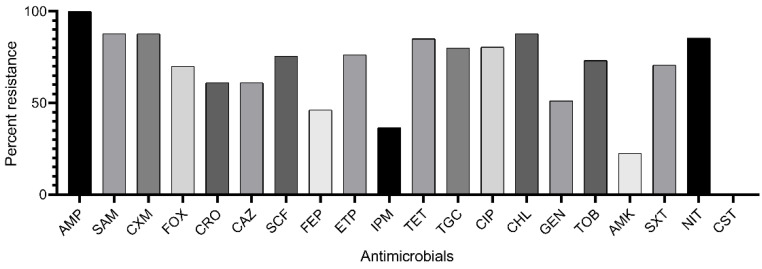
Rate of *K. pneumoniae* isolates resistant to antimicrobials: AMP, ampicillin; SAM, ampicillin-sulbactam; CXM, cefuroxime; FOX, cefoxitin; CRO, ceftriaxone; CAZ, ceftazidime; SCF, cefoperazone-sulbactam; FEP, cefepime; ETP, ertapenem; IPM, imipenem; TET, tetracycline; TGC, tigecycline; CIP, ciprofloxacin; CHL, chloramphenicol; GEN, gentamicin; TOB, tobramycin; AMK, amikacin; SXT, trimethoprim-sulfamethoxazole; NIT, nitrofurantoin; CST, colistin.

**Figure 2 antibiotics-10-00979-f002:**
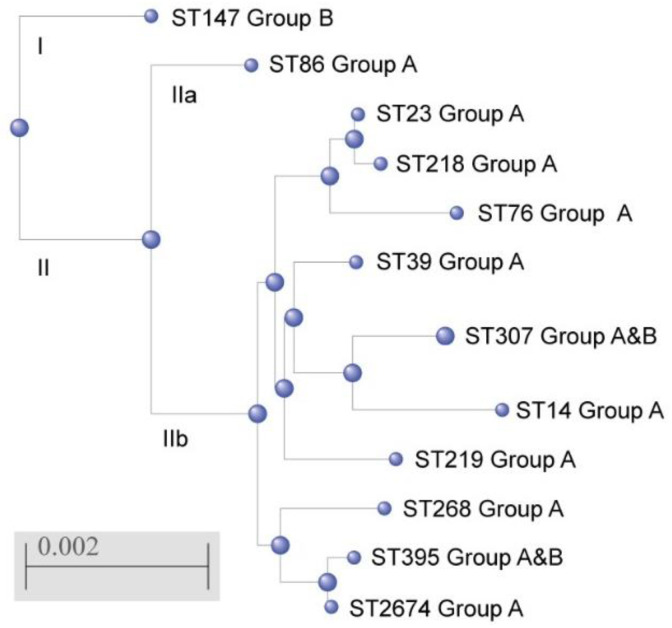
Phylogenetic tree of *K. pneumoniae* sequence types identified in the study generated by using a web resource NCBI “Blastn” and “Blast Tree View”, based on combined gene sequences of MLST profiles.

**Table 1 antibiotics-10-00979-t001:** Clinical data of the patients: underlying disease, infections, antimicrobial treatment, and outcomes.

Patient	Neurosurgery Disease	Infection	Antimicrobial Treatment	Outcome
Group A—patients with a pronounced systemic inflammatory response				
A	Hemorrhagic stroke.	Meningitis, abscess, and hemorrhage in the right frontal lobe of the brain, encephalitis, ventriculitis, epidural empyema, erythematous maculae and vesicles	Meropenem Tigecycline	Discharged
D	Multiple cerebral aneurysms, subarachnoid hemorrhage into the ventricular system.	Systemic inflammatory response, sepsis, septic shock, intracerebral hematoma	Cefoperazone-sulbactam Meropenem Colistin (inhalation)	Died
F	Rupture of aneurysm of the anterior cerebral artery, subarachnoid hemorrhage, ventriculoperitoneal shunt.	Meningitis, long-term (4 weeks) eradication of Klebsiella from cerebrospinal fluid, erythematous macules and vesicles	Meropenem Colistin (intrathecal)	Discharged
I	Multiple cerebral aneurysms, subarachnoid hemorrhage into the ventricular system, endovascular occlusion of the basilar artery aneurysm, basilar artery thrombosis.	Multiple focal hemorrhages of supra- and subtentorial localization, erythematous macules and vesicles	Meropenem Tigecycline Amikacin	Discharged
J	Odontoid process fracture of the C2 vertebra, dorsal fixation of the C1–C2 vertebrae.	Severe inflammatory reaction, increased markers of systemic inflammation, 50 mL of pus with blood was removed from the surgical wound.	Tigecycline, Meropenem, Ceftazidime-avibactam, Bacteriophage	Discharged
K	Complex defect of the base of the skull, spontaneous nasal liquorrhea, ventriculoperitoneal shunt.	Meningitis, thrombus in the pulmonary artery trunk, encephalitis, ventriculitis.	Meropenem Ciprofloxacin Amikacin (intrathecal)	Discharged
L	Mature teratoma of the chiasmal-sellar region, antitumor chemotherapy, removal of the teratoma	Meningitis, encephalitis, sepsis, severe septic reaction.	Meropenem	Discharged
M	Open penetrating traumatic brain injury, cerebral contusion, cerebral edema, subarachnoid hemorrhage.	Pneumonia with middle level markers of systemic inflammatory response.	Meropenem, Tigecycline Amikacin Colistin (inhalation)	Discharged
N	Large partially thrombosed cerebellar artery aneurysm, subarachnoid parenchymal hemorrhage, aneurysm clipping surgery.	Sepsis, septic reaction with markers of inflammation	Doripenem Amikacin Sulperason	Discharged
P	Gunshot wound to the skull and brain, Intraventricular hemorrhage, subdural hematoma of the frontal-parietal-temporal region	Klebsiella meningoencephalitis had a long persistent course, subarachnoid hemorrhage; markers of systemic inflammation were increased.	Meropenem Colistin	Discharged
Q	Closed traumatic brain injury, brain contusion with multiple hemorrhagic foci, subarachnoid hemorrhage, and occipital bone fracture.	Severe pneumonia, urinary infection, systemic inflammatory response, a rapid increase in inflammatory markers.	Tigecycline Meropenem	Died
R	Anaplastic ependymoma, tumor removal.	Fever, increased leukocytosis, markers of systemic inflammatory response, sepsis, meningitis	Meropenem, Ciprofloxacin	Discharged
S	Aneurysm of the right internal carotid artery, subarachnoid hemorrhage, aneurysm climax, ventricular drainage.	Meningoencephalitis, infection of the ventriculoperitoneal shunt, purulent masses in the lateral ventricles of the brain, pansinusitis.	Doripenem Colistin (inhalation) Colistin (intrathecal)	Died
Group B—patients without severe manifestations of systemic infections				
B	Rupture of middle cerebral artery aneurysms.	Urinary infection with increased markers of inflammation, fever.	Meropenem Tigecycline	Discharged
C	Ruptured aneurysm of the internal carotid artery.	Pansinusitis, no markers of inflammation.	Cefoperazone-sulbactam Meropenem	Discharged
E	Pilocytic astrocytoma, tumor biopsy, ventriculoperitoneal shunt.	Urinary infection without systemic markers of inflammation.	Amoxicillin-clavulanic acid	Discharged
G	Multiple metastases of kidney cancer in the ventricles of the brain, tumor removal.	Pneumonia with low markers of systemic inflammatory response. Died of an underlying disease.	Cefoperazone-sulbactam Meropenem	Died
H	Craniopharengioma, tumor removal.	Pneumonia without systemic markers of inflammation.	Meropenem Tigecycline Amikacin (intrathecal)	Discharged
O	Closed traumatic brain injury, subdural hematoma, cerebral edema, subarachnoid hemorrhage.	There was no septic reaction; markers of inflammation did not increase.	Meropenem, Tigecycline Colistin (inhalation)	Discharged
T	Arteriovenous malformation of cerebral vessels.	Tracheobronchitis, bacteriuria, leukocyturia.	Sulfamethoxazole/ Trimethoprim, Meropenem	Discharged

**Table 2 antibiotics-10-00979-t002:** Sources of isolation and antimicrobial resistance phenotypes of *K. pneumoniae* strains.

Strain	Date	Source	Patient	Resistance Phenotype	Resistance Category
Group A—strains associated with a pronounced systemic inflammatory response					
B-548/18	02-Apr-2018	trachea	D ^#^	BL^ACEI^, TET, QNL, CM, AMI, SUL, NIT	XDR
B-784/18	03-May-2018	urine	D ^#^	BL^ACEI^, TET, QNL, CM, AMI, SUL, NIT	XDR
B-1154/18	03-May-2018	blood	D ^#^	BL^ACE^, TET, QNL, CM, AMI, SUL, NIT	XDR
B-1396/18-2	25-May-2018	blood	J	BL^ACEI^, TET, QNL, CM, AMI, SUL, NIT	XDR
B-1618/18	27-Aug-2018	urine	Q ^#^	BL^ACE^, TET, QNL, CM, AMI, SUL, NIT	XDR
B-2035K/18-2	23-Jul-2018	CSF	P	BL^ACE^, TET, QNL, CM, AMI, SUL, NIT	XDR
B-2062/18	25-Jul-2018	CSF	P	BL^ACE^, TET, QNL, CM, AMI, SUL, NIT	XDR
B-2086/18	27-Jul-2018	CSF	P	BL^ACE^, TET, QNL, CM, AMI, SUL, NIT	XDR
B-968/18-1	30-May-2018	trachea	M	BL^ACE^, TET, QNL, CM, SUL, NIT	XDR
B-1120K/18	20-Jun-2018	trachea	M	BL^ACE^, TET, QNL, CM, SUL, NIT	XDR
B-2625/18	12-Dec-2018	trachea	S ^#^	BL^ACEI^, TET, QNL, CM, AMI, NIT	XDR
B-14/19	09-Jan-2019	CSF	S ^#^	BL^ACEI^, TET, QNL, CM, AMI, NIT	XDR
B-21/19	19-Jan-2019	trachea	S ^#^	BL^ACEI^, TET, QNL, CM, AMI, NIT	XDR
B-1398/18-1	25-May-2018	blood	L	BL^AC^, TET, CM, SUL	MDR
B-1406/18-1	25-May-2018	CSF	L	BL^AC^, TET, CM, SUL	MDR
B-1412/18-1	25-May-2018	blood	L	BL^AC^, TET, CM, SUL	MDR
B-1230/18-1	10-May-2018	CSF	K	BL^A^, CM, NIT	MDR
B-849/18-2	11-May-2018	trachea	K	BL^A^, CM, NIT	MDR
B-1207/18	03-Jul-2018	blood	N	BL^AC^	R
B-1636/18	30-Aug-2018	urine	R	BL^A^	R
B-2523/18	30-Aug-2018	blood	R	BL^A^	R
B-3002K/17 *	30-Oct-2017	CSF	A	BL^ACE^, TET, QNL, CM, AMI, SUL, NIT	XDR
B-3060K/17 *	03-Nov-2017	CSF	A	BL^ACE^, TET, QNL, CM, AMI, SUL, NIT	XDR
B-2016K/17 *	14-Nov-2017	BA	A	BL^ACE^, TET, QNL, CM, AMI, SUL, NIT	XDR
B-3299/17 *	24-Nov-2017	CSF	A	BL^ACE^, TET, QNL, CM, AMI, SUL, NIT	XDR
B-1040/18-1 *	09-Jun-2018	trachea	I	BL^AE^, QNL, CM, AMI, SUL, NIT	XDR
B-792/18 *	03-May-2018	trachea	I	BL^AE^, QNL, CM, NIT	MDR
B-853/18-1 *	14-May-2018	trachea	I	BL^ACEI^, QNL, NIT	MDR
B-1214/18-2 *	08-May-2018	CSF	F	BL^AC^, TET, QNL, SUL	MDR
Group B—strains not associated with severe manifestations of systemic infections					
B-789/18-1	03-May-2018	urine	E	BL^ACE^, TET, QNL, AMI, SUL, NIT	XDR
B-851/18-1	14-May-2018	urine	E	BL^ACE^, TET, QNL, AMI, SUL, NIT	XDR
B-790/18-1	03-May-2018	trachea	G ^#^	BL^ACEI^, QNL, CM, AMI, SUL, NIT	XDR
B-823/18-1	07-May-2018	trachea	G ^#^	BL^ACEI^, QNL, CM, AMI, SUL, NIT	XDR
B-702/18	20-Apr-2018	IS	C	BL^ACEI^, TET, QNL, AMI, SUL, NIT	XDR
B-771/18	28-Apr-2018	urine	B	BL^AEI^, QNL, CM, AMI, SUL, NIT	XDR
B-102/19	21-Jan-2019	urine	T	BL^ACEI^, TET, QNL, CM, AMI, SUL, NIT	XDR
B-543/18	02-Apr-2018	CSF	H	BL^ACEI^, TET, QNL, CM, AMI, SUL, NIT	XDR
B-587/18	09-Apr-2018	trachea	H	BL^ACEI^, QNL, CM, AMI, SUL, NIT	XDR
B-775/18-1	03-May-2018	trachea	H	BL^AE^, QNL, CM, AMI, SUL, NIT	XDR
B-691/18-4	19-Apr-2018	trachea	H	BL^ACE^, QNL, CM, NIT	MDR
B-1363/18-1	24-Jul-2018	trachea	O	BL^ACEI^, QNL, AMI, NIT	MDR

Abbreviations: *, strains associated with systemic manifestations of vasculitis and microcirculation disorders; CSF, cerebrospinal fluid; BA, brain abscess; IS, intracranial sinus; BL^ACEI^, beta-lactams (ampicillin, cephalosporins, ertapenem, imipenem); TET, tetracyclines; QNL, fluoroquinolones; CM, chloramphenicol; AMI, aminoglycosides; SUL, sulfonamides; NIT, nitrofurans; R, resistant; MDR, multidrug resistant; XDR, extremely drug resistant; ^#^, patient died.

**Table 3 antibiotics-10-00979-t003:** Molecular-genetic characteristics of *K. pneumoniae* strains.

Strain	Beta-Lactamase Genes	Int	Virulence Genes	ST	Capsular Type
Group A—strains associated with a pronounced systemic inflammatory response					
B-548/18	*bla*_SHV_, *bla*_CTX-M_, *bla*_TEM_, *bla*_OXA-48_		*uge, wabG, kfu, fimH, allS*	14	K2
B-784/18	*bla*_SHV_, *bla*_CTX-M_, *bla*_TEM_, *bla*_OXA-48_	*int1,2*	*uge, wabG, kfu, fimH, allS*	14	K2
B-1154/18	*bla*_SHV_, *bla*_CTX-M_, *bla*_TEM_, *bla*_OXA-48_	*int1*	*uge, wabG, kfu, fimH, allS*	14	K2
B-1396/18-2	*bla*_SHV_, *bla*_CTX-M_, *bla*_TEM_, *bla*_OXA-48_		*uge, wabG, fimH, allS*	2674	K47
B-1618/18	*bla*_SHV_, *bla*_CTX-M_, *bla*_TEM_		*rmpA, iroN, iroD, uge, wabG, fimH, allS*	268	K20
B-2035K/18-2	*bla*_SHV_, *bla*_CTX-M_, *bla*_TEM_	*int1*	*wabG, fimH, allS*	23	K57
B-2062/18	*bla*_SHV_, *bla*_CTX-M_, *bla*_TEM_		*wabG, fimH, allS*	23	K57
B-2086/18	*bla*_SHV_, *bla*_CTX-M_, *bla*_TEM_		*wabG, fimH, allS*	23	K57
B-968/18-1	*bla*_SHV_, *bla*_CTX-M_, *bla*_OXA-48_		*rmpA, wabG, fimH, allS*	23	K57
B-1120K/18	*bla*_SHV_, *bla*_CTX-M_, *bla*_TEM_, *bla*_OXA-48_		*rmpA, wabG, fimH, allS*	23	K57
B-2625/18	*bla*_SHV_, *bla*_CTX-M_, *bla*_TEM_, *bla*_OXA-48_		*uge, wabG, fimH, allS*	39	K23
B-14/19	*bla*_SHV_, *bla*_CTX-M_, *bla*_TEM_, *bla*_OXA-48_, *bla*_KPC_		*uge, wabG, fimH, allS*	39	K23
B-21/19	*bla*_SHV_, *bla*_CTX-M_, *bla*_TEM_, *bla*_OXA-48_		*uge, wabG, fimH, allS*	39	K23
B-1398/18-1	*bla*_SHV_, *bla*_CTX-M_		*uge, wabG, kfu, fimH, allS*	219	KL125/114
B-1406/18-1	*bla*_SHV_, *bla*_CTX-M_	*int1*	*uge, wabG, kfu, fimH, allS*	219	KL125/114
B-1412/18-1	*bla*_SHV_, *bla*_CTX-M_	*int1*	*uge, wabG, kfu, fimH, allS*	219	KL125/114
B-1230/18-1	*bla* _SHV_		*rmpA, iroN, iroD, wabG, fimH, allS*	218	K57
B-849/18-2	*bla* _SHV_		*rmpA, iroN, iroD, wabG, fimH, allS*	218	K57
B-1207/18	*bla*_SHV_, *bla*_CTX-M_		*rmpA, uge, wabG, fimH, allS*	76	K23
B-1636/18	*bla* _SHV_		*rmpA, iroN, iroD, uge, wabG, fimH, allS*	86	K2
B-2523/18	*bla* _SHV_		*rmpA, iroN, iroD, uge, wabG, fimH, allS*	86	K2
B-3002K/17 *	*bla*_SHV_, *bla*_CTX-M_, *bla*_TEM_		*uge, wabG, fimH, allS*	395	K2
B-3060K/17 *	*bla*_SHV_, *bla*_CTX-M_, *bla*_TEM_		*uge, wabG, fimH, allS*	395	K2
B-2016K/17 *	*bla*_SHV_, *bla*_CTX-M_, *bla*_TEM_		*uge, wabG, fimH, allS*	395	K2
B-3299/17 *	*bla*_SHV_, *bla*_CTX-M_, *bla*_TEM_		*uge, wabG, fimH, allS*	395	K2
B-1040/18-1 *	*bla*_SHV_, *bla*_OXA-48_		*uge, wabG, fimH, allS*	395	K2
B-792/18 *	*bla*_SHV_, *bla*_OXA-48_	*int1*	*uge, wabG, fimH, allS*	395	K2
B-853/18-1 *	*bla*_SHV_, *bla*_OXA-48_		*uge, wabG, fimH, allS*	395	K2
B-1214/18-2 *	*bla*_SHV_, *bla*_CTX-M_, *bla*_TEM_		*uge, wabG, fimH, allS*	307	KL102/149/155
Group B—strains not associated with severe manifestations of systemic infections					
B-789/18-1	*bla*_SHV_, *bla*_CTX-M_, *bla*_TEM_		*uge, wabG, fimH, allS*	307	KL102/149/155
B-851/18-1	*bla*_SHV_, *bla*_CTX-M_, *bla*_TEM_	*int1*	*uge, wabG, fimH, allS*	307	KL102/149/155
B-790/18-1	*bla*_SHV_, *bla*_OXA-48_	*int1*	*uge, wabG, fimH, allS*	395	K2
B-823/18-1	*bla*_SHV_, *bla*_OXA-48_		*uge, wabG, fimH, allS*	395	K2
B-702/18	*bla*_SHV_, *bla*_CTX-M_, *bla*_TEM_, *bla*_OXA-48_		*uge, wabG, fimH, allS*	395	K2
B-771/18	*bla*_SHV_, *bla*_CTX-M_, *bla*_OXA-48_	*int1*	*uge, wabG, fimH, allS*	395	K2
B-102/19	*bla*_SHV_, *bla*_OXA-48_	*int1*	*uge, wabG, fimH, allS*	395	KL39
B-543/18	*bla*_SHV_, *bla*_OXA-48_	*int1*	*uge, wabG, fimH, allS*	395	K2
B-587/18	*bla*_SHV_, *bla*_CTX-M_, *bla*_OXA-48_		*uge, wabG, fimH, allS*	395	K2
B-775/18-1	*bla*_SHV_, *bla*_OXA-48_		*uge, wabG, fimH, allS*	395	K2
B-691/18-4	*bla*_SHV_, *bla*_OXA-48_		*uge, wabG, fimH, allS*	395	K2
B-1363/18-1	*bla*_SHV_, *bla*_CTX-M_, *bla*_TEM_, *bla*_NDM_		*uge, wabG, fimH, allS*	147	K14/64

Abbreviations: *, strains associated with systemic manifestations of vasculitis and microcirculation disorders; *bla*_SHV_, *bla*_CTX-M_, *bla*_TEM_, *bla*_OXA-48_, *bla*_KPC-2_, *bla*_NDM_, beta-lactamase genes; *int1, int2,* integrase class 1 and 2 genes; *rmpA,* hypermucoid phenotype regulator gene; *iroN*, catecholate siderophore receptor gene; *iroD*, aerobactin esterase gene; *kfu*, ferric absorption system gene; *uge,* uridine diphosphate galacturonate-4-epimerase gene; *wabG*, glucosyltransferase gene; *fimH*, fimbria type I gene; *allS*, allantoin regulon; ST, sequence type.

**Table 4 antibiotics-10-00979-t004:** Molecular-genetic characteristics of *K. pneumoniae* isolates based on Whole-Genome Sequencing.

Isolate	B1396/18-2	B1120K/18	B14/19	B1406/18-1	B1230/18-1	B1207/18	B2523/18	B1214/18-2	B102/19
SCPM-O’s ID	B-9325	B-9326	B-9136	B;-9327	B-9138	B-9328	B-9220	B-9329	B-9137
ST/CT	ST2674/K47	ST23/K57	ST39/K23	ST219/K125	ST218/K57	ST76/K23	ST86/K2	ST307/K102	ST395/K39
Source	blood	trachea	CSF	CSF	CSF	Blood	blood	CSF	urine
Date	25-May-2018	20-Jun-2018	09-Jan-2019	25-May-2018	10-May-2018	03-Jul-2018	30-Aug-2018	08-May-2018	21-Jan-2019
BioSample ID	SAMN18679914	SAMN18679915	SAMN17885212	SAMN18679916	SAMN17885214	SAMN18679917	SAMN18679918	SAMN18679919	SAMN17885213
Read Archive	SRR14194623	SRR14194622	SRR13695927	SRR14194621	SRR13695925	SRR14194620	SRR14194625	SRR14194624	SRR13695926
GenBank	JAGRZJ000000000	JAGRZI000000000	JAFFJK000000000	JAGRZH000000000	JAFFJI000000000	JAGRZG000000000	JAGRZF000000000	JAGRZE000000000	JAFFJJ000000000
GC-content, %	56.76	57.07	56.76	57.26	57.29	56.9	57.39	57.38	57.24
Reads	1,461,812	1,268,077	4,656,796	1,031,631	904,468	920,438	659,438	932,557	939,118
Contigs, n	168	103	164	94	86	117	125	93	154
Genome, bp	5,777,126	5,581,990	5,855,608	5,407,960	5,422,578	5,724,981	5,430,135	5,434,336	5,494,443
Coverage, ×	58	49	196	44	44	37	24.8	40	42
N50 value, bp	128,340	136,532	247,851	219,102	126,261	162,973	71,482	179,463	106,056
Genes	5812	5540	5845	5309	5121	5711	5326	5291	5535
Antimicrobial resistance genetic determinants									
Beta-Lactams	*bla* _SHV-182_ *bla* _TEM-1B_ *bla* _CTX-M-15_ *bla* _OXA-1_ *bla* _OXA-48_	*bla* _SHV-33_ *bla* _TEM-1B_ *bla* _CTX-M-15_	*bla* _SHV-40_ *bla* _TEM-1B_ *bla* _CTX-M-15_ *bla* _OXA-1_ *bla* _OXA-48_ *bla* _KPC-2_	*bla* _SHV-26_ *bla* _CTX-M-15_	*bla* _SHV-33_	*bla* _SHV-59_ *bla* _CTX-M-15_	*bla* _SHV-28_	*bla* _SHV-28_ *bla* _TEM-1B_ *bla* _CTX-M-15_ *bla* _OXA-1_	*bla* _SHV-182_ *, bla* _OXA-48_
Aminoglycosides	*ant(2”)-Ia* *aac(6’)-Ib-cr* *aadA1*	*aph(6)-Id* *rmtB* *aph(3”)-Ib*	*ant(2′’)-Ia* *aac(6′)-Ib-cr* *aadA1*	*aadA2* *aph(3”)-Ib*	*aph(3’)-VIa* *aph(6)-Id* *aph(3′’)-Ib*	*aph(6)-Id* *aph(3”)-Ib*		*aac(6′)-Ib-cr* *aph(6)-Id* *aph(3”)-Ib*	*aadA5* *armA*
Fosfomycin	*fosA*	*fosA*	*fosA*	*fosA*	*fosA*	*fosA*	*fosA*	*fosA*	*fosA*
Phenicols	*catA1* *catB3*	*catA2*	*catA1* *catB3*	*catA2*	*catA3*			*catB3*	*catA1*
Quinolones	*aac(6’)-Ib-cr* *qnrS1* *oqxA* *oqxB*	*qnrS1* *qepA1* *oqxA* *oqxB*	*qnrS1* *aac(6’)-Ib-cr*	*qnrS1* *oqxA* *oqxB*			*oqxA* *oqxB*	*aac(6′)-Ib-cr* *qnrB1* *oqxA* *oqxB*	
Sulfonamides	*sul1*	*sul2*	*sul1*	*sul1* *sul2*				*sul2*	*sul1*
Tetracyclines	*tet*	*tet*		*tet*				*tet*	*tet*
Trimethoprim	*dfrA1*			*dfrA12*				*dfrA14*	*dfrA17* *dfrA1*
Macrolides				*mph*					
Efflux	*acr env fis mar oqx ram rar rob sdi sox*	*acr env fis mar oqx rar rob sdi sox*	*acr env fis mar oqx ram rar rob sdi sox*	*acr env fis mar oqx ram rar rob sdi sox*	*acr env fis mar oqx ram rar rob sdi sox*	*acr env fis mar oqx ram ram rob sdi sox*	*acr env fis mar oqx ram rar rob sdi sox*	*acr env fis mar oqx ram rar rob sdi sox*	*acr env fis mar oqx rar rob sdi sox*
Virulence genetic determinants (locuses)									
Type 3 adhesin	*mrk*	*mrk*	*mrk*	*mrk*	*mrk*	*mrk*	*mrk*	*mrk*	*mrk*
Yersiniabactin biosynthesis	*irp*	*irp*	*irp*		*irp*		*irp*	*irp*	*irp*
Yersiniabactin transcriptional regulator	*ybt*	*ybt*	*ybt*		*ybt*		*ybt*	*ybt*	*ybt*
Siderophore yersiniabactin receptor	*fyu*	*fyu*			*fyu*		*fyu*	*fyu*	*fyu*
Aerobactin siderophore synthesis	*iuc*	*iuc*	*iuc*		*iuc*		*iuc*		
Ferric aerobactin receptor	*iut*	*iut*	*iut*		*iut*		*iut*	*iut*	
Regulator of capsular polysaccharide synthesis					*kvg*		*kvg*		
Plasmids	IncFIB IncHI1B IncR	IncC IncFIA IncFII	IncFIB IncFII IncHI1B	IncFIB	IncM1	IncFIB	IncHI1B	IncFIB	IncM2 IncR

Abbreviations: *pco*, encoded copper (Cu) resistance; *pbr*, lead (Pb) resistance; *sil*, silver (Ag) resistance; *ars*, arsenic (As)resistance; *ter*, tellurium (Te) resistance.

## Data Availability

Following Whole genome sequences were deposited in the GenBank database: JAGRZJ000000000, JAGRZI000000000, JAFFJK000000000, JAGRZH000000000, JAFFJI000000000, JAGRZG000000000, JAGRZF000000000, JAGRZE000000000, and JAFFJJ000000000.

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
