# Peer review of "Multidrug-Resistant Klebsiella pneumoniae Causing Severe Infections in the Neuro-ICU"

_antibiotics, 2021, doi:10.3390/antibiotics10080979_

Round 1

Reviewer 1 Report

In this manuscript Fursova et al. characterized Klebsiella pneumoniae (Kp) strains isolated from patients in a neuro-ICU unit in a hospital in Moscow. The Kp strains were isolated from the following types of health-associated infections: bloodstream, respiratory, urinary tract, and ventriculitis and meningitis. These Kp strains were analyzed for their antimicrobial susceptibility, virulence genes, antibiotic resistance gene, sequence/capsule types and further characterized through whole genome sequencing. In general, the experimental methods of this study are well described and the results of this study is straightforward. The manuscript should be accepted with the following requested modifications:

  1. Line 91-94: The authors mention that group A isolates were obtained from patients D, S and L. However, table 2 shows main other patients. This section needs to be explained more clearly.
  2. Line 102: Please correct ‘tree’ to three.
  3. Line 137: Please correct ‘tree’ to three.
  4. Line 154: There is no description of which isolates were selected from groups A and B. Please describe the isolates and the rationale for how the 8 isolates from group A were selected for whole genome sequencing.
  5. The authors mention the general virulence factors in group A strain but do not really discuss as how these virulence factors can increase pathogenicity of the strains.
  6. The discussion section contains a lot of repetition of the results. This discussion should offer some more broader outlook of how the observations of these studies can be employed in future studies or to improve treatment.

Author Response

We are grateful to Reviewer 1 for the positive assessment of our manuscript. We can answer the questions as follows:

Point 1: Line 91-94: The authors mention that group A isolates were obtained from patients D, S and L. However, table 2 shows main other patients. This section needs to be explained more clearly.

Response 1: According to suggestions of Reviewer 1 we added information in the paragraph: “Depending on clinical manifestations, isolates were combined into two groups. Group A included 29 isolates collected from 13 patients (D, J, Q, P, M, S, L, K, N, R, A, I, and F) with the pronounced systemic inflammatory response (SIRS). Symptoms have been reported for the patients: fever > 38 °C, leukocytosis, increased markers of systemic inflammation, multiple hemorrhages, thrombosis, or high risk of thromboembolism. Clinical forms have been detected: meningitis after the neurosurgical intervention, pansinusitis, pneumoniae, sepsis and septic shock, ventriculitis, and brain abscess. In this group, seven isolates (B-548/18, B-784/18, B-1154/18, B-1618/18, B-2625/18, B-14/19, and B-21/19) were associated with fatal incomes collected from the patients D, Q, and S died due to increased symptoms of inflammation, sepsis, septic shock, meningoencephalitis associated with K. pneumoniae infection. Other eight isolates (B-3002K/17, B-3060K/17, B-2016K/17, B-3299/17, B-1040/18-1, B-792/18, B-853/18-1, and B-1214/18-2) of Group A were associated with manifestations of vasculitis and microcirculation disorders (patients A, I, and F) (Figure 1).”

Point 2:  Line 102: Please correct ‘tree’ to three. Line 137: Please correct ‘tree’ to three.

Response 2: It was done.

Point 3: Line 154: There is no description of which isolates were selected from groups A and B. Please describe the isolates and the rationale for how the 8 isolates from group A were selected for whole genome sequencing.

Response 3: The reason for the WGS of the 8 isolates from Group A was their belonging to 8 combinations of sequence types / capsular types. Addition information was presented in the text: “Whole-genome sequencing was done for nine isolates including eight isolates of Group A belonging to sequence types / capsular types ST2674/K47, ST23/K57, ST39/K23, ST219/K125, ST218/K57, ST76/K23, ST86/K2, and ST307/K102 and one isolate of Group B belonging to ST395/K39.”

Point 4: The authors mention the general virulence factors in group A strain but do not really discuss as how these virulence factors can increase pathogenicity of the strains.

Response 4: According to the Reviewer’s suggestion, we added discussion about contribution to virulence the virulence factors coding by rmpA, iroN, iroD, and kfu genes: “It was reported previously that overexpression of rmpA could enhance the virulence of K. pneumoniae isolates in the mouse model []”; “It is known that the majority of pathogenic bacteria including K. pneumoniae possess the iron-acquisition system with a higher affinity for iron than the host, which serves as one of the strategies for increasing bacterial virulence []”; “Kfu was shown to be a potential virulence factor in the intragastrical murine model, which indicates that kfu might contribute to intestinal colonization”.

Point 5:The discussion section contains a lot of repetition of the results. This discussion should offer some more broader outlook of how the observations of these studies can be employed in future studies or to improve treatment.

Response 5: The “Discussion” section was expanded and modified.

Reviewer 2 Report

Authors aimed to examine the phenotypes, genotypes, and genetic relatedness of K. pneumoniae isolates recovered from the patients of neuro-ICU and evaluated the sequence types associated with severe infections in 2017-2019. The subject is relevant and and it should be of interest for the readers of the Journal. However, shortcomings must be corrected and further analysis performed. I am listing below points that  should be addressed by Authors:

AUTHORS: “Today, 13 more than 100 K. pneumoniae independent phylogenetic lineages or ‘clones’ have been 14 discovered”

> Provide reference

AUTHORS: “Classical K. pneumoniae (cKP) characterized in a great possibility to accumulate 23 resistance mechanisms to antimicrobial”

“The global spread of ‘convergent’ K. pneumoniae strains that 34 combined pathogenic potentials of cKP”

> Define cKP

> Figure 1. It is not necessary.

> The tables could be reorganized and combined to show only relevant characteristics

> Authors could display a graphic representation about antimicrobial resistance results.

> Figure 2. Low resolution.

> Authors should determine Minimum inhibitory concentrations (MICs) of the antimicrobials.

> Polymyxin susceptibility tests  should be performed.

> For a  molecular characterization, it should be interesting to analyze the complete genetic structures surrounding the antimicrobial resistance genes.

AUTHORS: “AM resistance genes, STs, plasmids, and restriction- 347 modification systems were identified using the web resource of the Center for Genomic 348 Epidemiology (http://www.genomicepidemiology.org/).”

> I suggest the authors look for genes associated with resistance by other means. For example: Identification of acquired antibiotic resistance genes using Kmers Kmeresistance.

> For gene prediction, please specify which were the select thresholds for minimum % identity and select minimum % coverage used.

Author Response

We are grateful to Reviewer 2 for the positive assessment of our manuscript. We can answer the questions as follows:

Point 1: AUTHORS: “Today, more than 100 K. pneumoniae independent phylogenetic lineages or ‘clones’ have been discovered” > Provide reference

Response 1: The sentence has been corrected as follows: “Today, according to BIGSDB Institute Pasteur database (https://bigsdb.pasteur.fr/), 5797 K. pneumoniae sequence types and 711 capsular types have been discovered”

Point 2: AUTHORS: “Classical K. pneumoniae (cKP) characterized in a great possibility to accumulate resistance mechanisms to antimicrobial” “The global spread of ‘convergent’ K. pneumoniae strains that combined pathogenic potentials of cKP” > Define cKP

Response 2: According to the suggestion, we corrected the phrase: “Classical K. pneumoniae (cKP) are the common healthcare-associated pathogens caused nosocomial infections in immunocompromised patients and characterized in a great possibility to accumulate resistance mechanisms to antimicrobials; multidrug-resistant (MDR), extensive drug-resistant (XDR), and pan drug-resistant (PDR) strains have been described [8].”

Point 3:> Figure 1. It is not necessary.

Response 3: We removed Figure 1 from the text.

Point 4: > The tables could be reorganized and combined to show only relevant characteristics

Response 4: It was done.

Point 5:> Authors could display a graphic representation about antimicrobial resistance results.

Response 5: The results were presented in Figure 1. Rate of K. pneumoniae isolates resistant to antibacterials.

Point 6: > Figure 2. Low resolution.

Response 6: Figure 2 was modified.

Point 7: > Authors should determine Minimum inhibitory concentrations (MICs) of the antimicrobials.

Response 7: Antimicrobial MICs were presented in Table S1.

Point 8: > Polymyxin susceptibility tests  should be performed.

Response 8: Susceptibility to colistin was performed using Vitek 2 instrument. The data were added to Figure 1 and Table S1.

Point 9: > For a molecular characterization, it should be interesting to analyze the complete genetic structures surrounding the antimicrobial resistance genes.

Response 9: Accordingly to the Reviewer’s suggestion we analyzed genetic structures surrounding carbapenemase genes blaKPC‑2 and blaOXA-48 and cefalosporinase gene blaCTX-M-15. The following information was added to the “Results” section: “The genetic environments of carbapenemase genes blaKPC‑2 and blaOXA-48 and cefalosporinase gene blaCTX-M-15 were similar to previously described for these genes. The gene coding a putative transposition helper protein ISKpn7 was identified upstream blaKPC‑2 gene, and ISKpn6 transposase gene – downstream, that was similar to the genetic structure in the plasmid pBK32179 (JX430448). The environment for blaOXA-48 genes contained upstream the transcriptional regulator LysR gene, and the transposase tir gene downstream – like in the plasmid pOXA-48 (JN626286). The surrounding genetic conditions for blaCTX-M-15 genes in clinical isolates were likewise to the plasmid pKp848CTX (NC_024992): ISEcp1 family transposase upstream, and gene coding WbuC family cupin fold metalloprotein downstream.”

Point 10: AUTHORS: “AM resistance genes, STs, plasmids, and restriction- modification systems were identified using the web resource of the Center for Genomic Epidemiology (http://www.genomicepidemiology.org/).”

> I suggest the authors look for genes associated with resistance by other means. For example: Identification of acquired antibiotic resistance genes using Kmers Kmeresistance.

> For gene prediction, please specify which were the select thresholds for minimum % identity and select minimum % coverage used.

Response 10: Thank you for your careful reading and suggestion for clarification. We used the KmerResistance 2.2 service of the Center for Genomic Epidemiology. We added information in the “Material and Methods” section: “AM resistance genes, STs, and plasmids were identified using the web resources of the Center for Genomic Epidemiology: ResFinder, KmerResistance (90 % identity threshold and 10 % threshold for depth corr.), MLST, and PlasmidFinder (95 % threshold for minimum identity and 60 % minimum coverage) (http://www.genomicepidemiology.org/)”